# A Study in Scatter: Investigating Low-Contrast Image Contents Outside the X-Ray Collimation

**Abstract.** Fluoroscopy is a widely used modality that provides vision to surgeons in minimally invasive surgery, but inherently raises concerns about radiation exposure. Collimation is a technique to reduce exposure by narrowing the radiation to a smaller area, with the trade-off of field-of-view limitations. However, the constraint the collimator shutters form for the x-ray beam is not absolute. Due to the non-ideal properties of the x-ray imaging and collimation process, small amounts of radiation are detectable outside the collimated area. This is a source of additional information, freely available as a byproduct of the imaging process, yet currently left unregarded. We explore whether this information can be used to provide additional knowledge about the surgical scene. In particular, we investigate whether it can be used to detect and visualize anatomical landmarks and surgical devices outside of the collimated area. We discuss the origins of this phenomenon, and perform experiments to evaluate its properties under different x-ray source parameters. Using anthropomorphic phantoms and a set of surgical guidewires, we investigate how well and under which conditions different landmarks and devices can be visualized with the proposed concept. We hope this work can open a path to provide additional information to interventional radiologists, while making use of every bit of radiation the patient is exposed to.

**Keywords:** X-Ray Imaging · Minimally Invasive Surgery · Interventional Radiology · Collimation

## 1 Introduction

In minimally invasive interventions, surgeons use fluoroscopy to visualize anatomical landmarks and surgical instruments inside the human body. As fluoroscopy is a live stream of X-ray images, it subjects both patient and clinical staff to radiation. A way to reduce the radiation dose is to narrow down the x-ray beam, so that it irradiates only the anatomical region of interest. This *collimation* is done by sliding metallic collimator shutters into the path between X-ray source and patient. The shutters form a window that constrains the emitted radiation to the area of interest, blocking rays that would irradiate the rest of the patient (Fig. 1a). Collimation decreases the radiation dose, with the trade-off of limiting the surgeon's field of view. In an idealized scenario, collimation would lead to

a part of the X-ray detector being fully irradiated and the rest of the detector receiving no radiation at all, creating an image as in Fig. 1c. In this work, we explore subtle information contained in the parts of the detector that are shielded by the collimator shutters - regions that would not receive any radiation based on the idealized scenario described above - and investigate if such information can be used to alleviate the field of view limitation of collimation. We discuss the origins of this information and investigate its properties in a mobile imaging robot. We also investigate how bones and surgical tools are represented in this information, and explore if we can use it to reconstruct their shape and location. To the best of our knowledge, we are the first to explore the potential this phenomenon could hold for reconstruction outside the collimated field of view.

## 2   Related works

During a procedure, the surgeon needs to see both anatomical landmarks, as well as the medical devices they currently manipulate inside the body. While static collimation to the overall anatomical region of interest can reduce radiation, stronger dose reduction is possible. Previous works proposed to collimate as narrowly as possible at each point in time, providing the surgeon with just enough information to perform the current step of the procedure. This is done by automatically finding the current region of interest (e.g. the location of a navigated surgical instrument) and dynamically changing the collimated image interval to encompass it [4]. While their approach reduced radiation, they recognized the risk that some information could be lost due to the reduced field of view from collimation. Therefore, they merged the constantly updated collimated image with uncollimated images taken at a lower framerate, opening up the collimator shutters once per second. This updates the peripheral image regions at a lower frequency than the central area - a compromise to retain full-field-of-view context while reducing radiation. A successful clinical use case of this method was shown as well [1]. Instead of temporally merging background and collimated frames, another work proposed to spatially modulate the emitted X-ray intensity by mounting a region-of-interest (ROI) attenuator, made of a copper plate with a central hole, in front of the X-ray source [5]. While the central beam is unimpeded, radiation to peripheral regions is reduced by around 80% through the copper plate (different from collimator shutters that block almost all radiation from traversing). This provides the surgeon with a central image region benefiting from full X-ray exposure and sharp contrast and a peripheral image with lower contrast (and reduced radiation dose) to retain full-field of view.

   Both temporal and spatial approaches aim at dose reduction while preserving full field of view. Compared to these works, we are interested in whether we can reconstruct information from regions outside the collimated area without having to un- and recollimate (as in [4]) and without modifying the device as in [5], using information naturally emerging from the image formation process. We are specifically interested in features in the regions of collimated images that correspond to parts of the detector shielded by the collimator shutters. A collimated X-ray

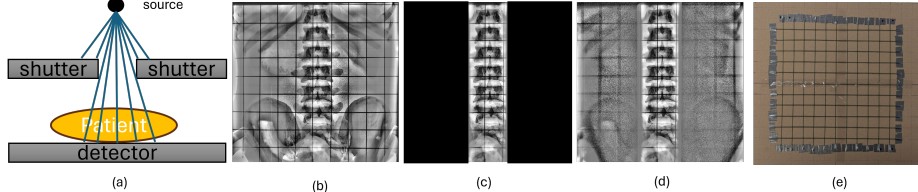

**Fig. 1.** (a) Collimated X-ray imaging; (b) uncollimated image of phantom #2 with guidewire grid on top; (c) idealized (artificial) collimated image, (d) a real collimated image with bone structures appearing in the regions outside the collimated area; (e) manufactured wire grid. (b), (c) and (d) postprocessed using histogram normalization

image consists of two parts: the central, *collimated region* between the shutters and the *shielded region* outside the collimated area. The collimated region stems from the parts of the detector that directly receive radiation from the source, geometrically determined by a theoretical point source irradiating the X-rays and the window formed by the collimator shutters. Under the idealized assumptions of a) a point radiation source, b) collimator shutters entirely blocking incoming radiation, and c) no scattering events, these are the only parts of the detector receiving radiation from the source, creating an image as in Fig. 1c. The collimator shutters block direct radiation from reaching the shielded regions. However, the X-ray image generation process is non-ideal. The X-ray source is not a point source; it has a finite focal spot distribution, creating additional off-focal spot radiation [2]. Also, scattering events occur inside the X-ray tube, the collimator, and the patient. These effects form a distribution of sources. Most interestingly, this leads to image information outside the collimated regions. A reconstruction of such information in the shielded regions using local histogram equalization [6] is visualized in Fig. 1d. Strongly attenuating structures, such as bones, are visible in regions outside of the collimated area.

## 3 Materials and Methods

We experimentally confirmed and evaluated this with Loop-X, a mobile imaging robot. We collected native, raw detector images (1440x1440). Collimator shutters of Loop-X are proven radio-intransparent (99.99% absorption), so our results are not due to X-rays penetrating the collimator. We were interested in visibility of bones or medical devices in the shielded regions. To investigate device visibility across different image parts all at once, we used Amplatz Extra Stiff Guidewires (diameter 0.89mm, Cook Medical), routinely used endovascular surgical instruments. We cut the wires and manufactured a 30x30 cm grid on cardboard, with 2.5cm spacing (Fig. 1e). As shielded regions only receive radiation with an intensity far smaller than the main beam's, information in those regions is noisy and image features have poor contrast. We visualize this information using local histogram equalization (CLAHE [6], Filter Size 71x71, $\alpha=\beta=0.0$).

## 4    Experiments

We performed 3 sets of experiments to investigate different influencing factors.

*Open Beam Properties* We evaluated the raw image intensity distributions on the detector when the beam is collimated, but no object is placed between detector and source. We investigated the influence of several parameters on this distribution in the shielded regions. Geometric parameters were size, position and aspect ratio of the collimation box. Exposure parameters were tube current, voltage and focal spot size (Loop-X offers 0.3mm and 0.6mm). Tube current was 0.5mA or 1.0mA; voltage 80 or 120kV. This low current setting avoids overexposure while no object attenuates the beam. For each exposure setting, we tested a set of box sizes, either collimated to the center of the detector, or to the point opposite the focal spot of the source (where the central line of the X-ray beam hits the detector). For some settings we sampled more box positions and varied box anisotropy. We always took an uncollimated reference image. For each image, we recorded 12 frames of 84ms, for 1 second (1008 ms) exposure time.

*Wire Grid* To test how the results of the first experiment translate to visibility of medical devices, we took collimated images with a wire grid in the X-ray beam.

*Anthropomorphic Phantoms* We used two anthropomorphic X-ray torso phantoms (The Phantom Laboratory & Kyoto Kagaku) for our final experiment. Phantom #1 contains a human skeleton cast into a material with the same effective atomic number as human soft tissue. Phantom #2 consists of resins (radiological absorption and HU number comparable to the human body). We recorded collimated images with the wire grid on top of the phantoms, in different exposure and geometric parameterizations. Default source parameters were 120kV, 4.0mA, anterior-posterior (AP) orientation, large focal spot.

## 5    Results

As the information in the shielded regions is noisy, we present results obtained by averaging 12 frames (1 second of Fluoroscopy), unless stated otherwise.

*Open Beam Properties* We confirmed residual radiation in the shielded regions. To understand its magnitude, we divided each collimated image by an uncollimated reference image. Thereby, we obtained for each pixel the percentage of intensity recorded during collimation compared to the intensity without collimation. For two sets of square collimation boxes, the results are shown in Fig. 2. As expected due to the orientation of the rotating anode [2], we found that radiation in the shielded regions is much more prevalent in the horizontal axis. Only little residual radiation could be found in the vertical direction. With increasing box size, the irradiated area both grows wider and the intensity inside of it increases. For two different box positions (detector center and opposite focal spot), this pattern was comparable, though the exact shape shows some dependency on position. We present results from different x-ray source parameters in Fig. 3a. Lower tube voltage (80 kV) showed increased radiation in the shielded areas compared to 120 kV across the whole detector. Focal spot size impacted the

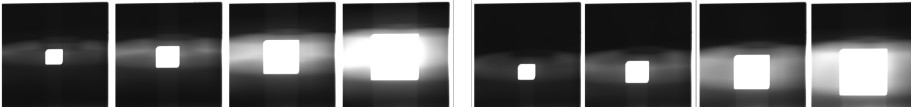

**Fig. 2.** Irradiation patterns of collimation boxes, at 120kV, 0.5mA with a large focal spot. Left cluster is centered to detector center, right is opposite to focal spot. Each pixel represents the percentage of radiation the respective detector cell recorded compared to an uncollimated image. Images are cropped to [0.0, 0.05] for visibility (white detector regions still recorded $\geq 5\%$ of the uncollimated radiation intensity despite being behind collimator shutters). Central white squares are the collimated area, outside regions indicate radiation in shielded areas

irradiation pattern locally, but did not overall increase or decrease the amount of radiation in the shielded regions. Results for anisotropic boxes spanning the whole detector width or height are shown in Fig. 3b. Column-wise collimation showed prevalence of irradiation in the upper half of the detector, and that radiation extended further toward the left than to the right when collimating off-center. Row-wise collimation confirmed that vertically, not much radiation is found. Results for smaller anisotropic boxes are in Fig. 3c. Extending boxes vertically lead to a larger shielded area being affected. Horizontal extension increased the intensity in the affected areas. Vertical box position also slightly influenceed the vertical irradiation pattern, extending away from the center (Fig. 3d).

*Wire Grid* The grid could successfully be visualized in the shielded regions, with characteristics similar to the open beam experiments, most notably the horizontal prevalence. Interestingly, while larger collimated boxes did increase the irradiated area as expected, they also increasingly blurred the wires (Fig. 4). Additionally, a geometric deformation was observed: the reconstructed wires were shifted compared to their uncollimated position (see overlay in Fig. 4e). We attribute this to the fact that the information in shielded regions comes from off-focal and scattered radiation. Deformation grew with distance from center.

*Anthropomorphic Phantom* In the anthropomorphic phantoms, we investigated the interplay of tissue properties and visibility of bones and devices. In Phantom #1, in AP recording, we could successfully visualize both wires and certain bones in the shielded regions (Fig. 5a, Images 1,2,4). Visibility depended on total attenuation in the path through the body, with e.g. ribs being easier to recover in front of lungs (Fig. 5a, region R1 in red) than in front of abdominal tissue (Fig. 5a, region R2). Similarly, wires were clearly visible in front of lungs and ribs, yet visibility decreased in front of soft tissue. Wires overlapping with the spine were barely noticeable (Fig. 5a, img. 5 & 6, R3). The finding from open beam experiments that radiation extends more toward the left than to the right for off-center collimation was also seen here (Fig 5a, img. 5 vs 6). A higher number of averaged frames visualized bones and wires more finely (Fig. 5a, img. 1 & 4), also when they suffer from reduced visibility due to attenuating tissue or bones. Smaller collimation boxes showed the same horizontal preference as in the open beam experiments, combined with the finding from the wire grid experiments

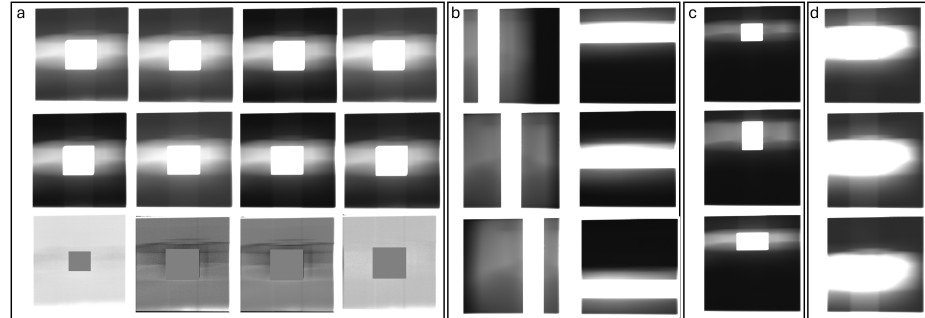

**Fig. 3.** (a) Comparison of tube settings. In each column, the first and second image are two different settings and the third is the difference between them. For visibility, images are cropped to a range of [0.0, 0.05] and difference images to [-0.01, 0.01]. In difference images, the center is set to 0.0 to provide a reference of regions with increased intensity (brighter than center) or decreased (darker). From left to right: 80kV vs 120kV tube voltage at (0.5mA-large FS); Large vs small focal spot at (80kV-0.5mA), Large vs small focal spot at (120 kV - 0.5mA), 0.5mA vs 1.0 mA at (80kV-large FS. (b) Images obtained with different column and row collimations (c) different aspect ratios (d) influence of box size on vertical collimation beam (images cropped to [0.0, 0.02]



**Fig. 4.** Wire with different collimation sizes, (a) uncollimated, (b) 5cm, (c) 12.5cm, (d) 22.5cm. (e) zoomed-in overlay of lower-left quarter of (a) and (b). CLAHE 11x11

that larger boxes blurred wires more than small ones (Fig. 5a, Image 3). In phantom #2, we found that higher tube voltage lead to sharper, more distinguishable image features for ribs, hips (Fig 5b, tube voltage section, red arrows) and wires. Lowering tube current from 4mA to 2mA reduced clarity (Fig 5b, tube current section). To evaluate wire sharpness and contrast, we calculated the wires' full width at half maximum (FWHM) and contrast-to-noise ratio (CNR). FWHM was computed on a gaussian ($\sigma = 1.0$) and median filtered (kernel 7) average of 100 rows (center and boundaries shown as yellow lines in Fig. 5b). CNR was calculated from an unfiltered patch consisting of the same 100 rows. The CNR signal level was calculated as mean of the region $\pm0.5$*FWHM around the center (peak) of the wire. CNR noise level and standard deviation were estimated from regions $\pm2.5$-$4.0$*FWHM left and right of the center. Mean wire FWHM and CNR are stated above the images in Fig. 5b. Comparing AP orientation of source and detector with PA orientation (Fig 5b, orientation section), we found the grid to be sharper in PA. This likely results from the grid being closer to the

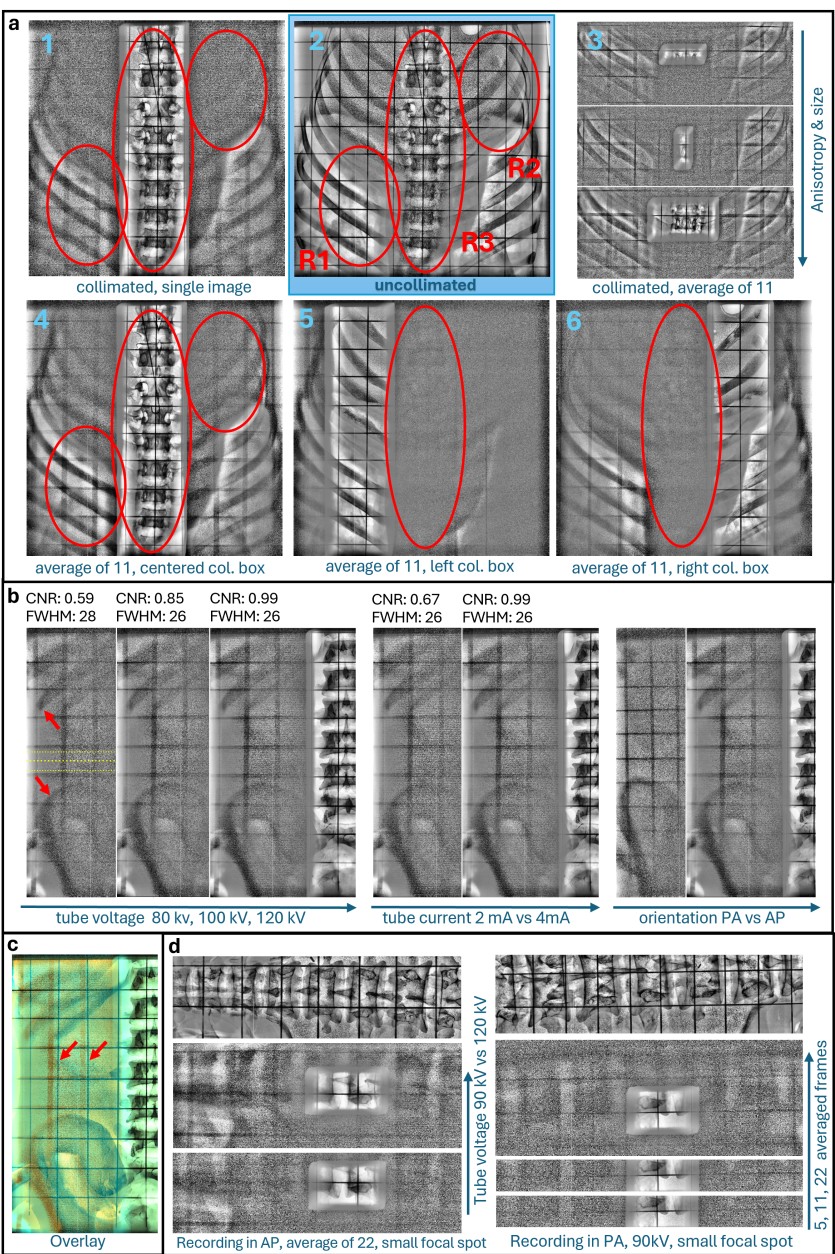

**Fig. 5.** Various phantom+grid recordings, all CLAHE filtered (a) phantom #1, analysing averaging, box position and size, (b) tube voltage, current and orientation in abdominal area of phantom #2 (uncollimated image in Fig. 1b) (c) overlay showing geometric deformation (d) cropped view of spine of phantom #2; phantom rotated to align with dominant axis of radiation, comparing orientation, averaging and voltage

detector in PA setting, leading to lower magnification and higher image clarity. Geometric deformation was again observed, mostly horizontal (Fig. 5c).

In some procedures, guidewires move in the abdominal aorta above the spine. Thus, we explored if we could visualize wires even in front of the spine. To align the horizontal irradiation pattern with the spine, we rotated the phantom by 90°. We chose a small collimation box, as would be the case if focusing on the tip of a guidewire, which as we know from above results also produces sharper features than a larger collimation box. We could visualize the wire in multiple scenarios. In AP orentation, for 120kV, the wire could be visualized, but visualization was poor at 90kV (Fig. 5d, section recording in AP). In PA, we could visualize the wire at 90 kV, with clarity increasing for more averaged frames (Fig. 5d, section recording in PA). For reference, a study in endovascular abdominal interventions reported 3.1 mA and 81kV as low-dose protocol [3], so our parameters of 2.0 to 4.0 mA and 80 to 120kV are in a reasonable range for minimally invasive surgery.

## 6    Conclusion

We introduced and explored the use of information contained in shielded regions of collimated images in order to reconstruct bones and surgical devices outside of the collimated area. This concept is of interest because in clinical settings, radiation protection and procedural safety are both crucial. We envision our approach integrated into a system that tracks and collimates to the tip of the device the surgeon currently manipulates, which is the most important part of the device they need to see. Narrow collimation to the tip would reduce patient dose, yet there are important warning signs the surgeon might miss - e.g. if outside the collimated area the device loops or kinks along the shaft. Also, if during advancement and tracking of a stent graft system over a wire, the end of the wire is unintentionally advanced (e.g. into the heart during TEVAR), the surgeon must know. Or if an already placed therapeutic device (e.g. stent graft) dislodges while he manipulates another device. Finally, instead of relying only on a small collimation window, reconstruction of bones could help to ensure that registration with a static model (2D contrast run or 3D CTA) stays accurate in case of patient motion, e.g. in PAD intervention with local anesthesia. Providing awareness about the surgical scene, while reducing radiation by narrow collimation, is thus a clinical setting that would benefit from our approach.

We view our work as an initial discussion and investigation into this subject, to introduce it to the scientific community. The results indicate that this concept has potential to alleviate some of the field-of-view limitations of collimated imaging while inducing no additional radiation to the patient. Yet, it is also clear that each potential application has to be carefully chosen, as spatial characteristics like the horizontal prevalence, interaction of devices with highly attenuating structures like the spine, and temporal considerations about averaging frames containing moving objects need to be considered. We hope this work inspires researchers and device manufacturers to explore the use of this concept and enables them to provide more information to the surgeon.

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
