# OpenReview forum: "A Study in Scatter: Investigating Low-Contrast Image Contents Outside the X-Ray Collimation"
_MICCAI.org/2025/Workshop/MSB_EMERGE — MSB EMERGE 2025 Oral_

### Official Review · Reviewer_G4Bo · 2025-07-02

**Recommendation:** 5
**Confidence:** 3

**Clarity:**

The paper is clear and well-written, with minor areas for improvement in clarity

**Feedback:**

Figure Captions: Please include the units for the windowing ranges in Figure 2 and all other relevant figure captions for clarity.

Section Title and Content: Section 5 should be renamed to “Results and Discussion”, and additional discussion of the findings should be incorporated to better contextualize the results.

Frame Averaging Inconsistency: There is an inconsistency regarding the number of frames averaged—12 frames are mentioned in the text, while Figure 5 states that 11 frames were used. Please ensure consistency throughout the paper.

Motion During Averaging: Consider discussing the potential impact of movement during frame averaging, as this could affect image quality and interpretation.

Collimation and Blurring: The observation that larger collimated boxes result in increased wire blurring should be discussed in more depth. Could this be attributed to greater contributions from undirected scatter or noise?

Wire Deformation: While the paper notes wire deformation, the origin of this deformation should also be addressed to clarify whether it is due to physical setup, imaging artifacts, or other factors.

Figure 4: Please indicate the zoomed-in region within images a–d to help the reader understand what is being magnified.

PA Grid Sharpness vs. Rib Contrast: Although the grid appears sharper in the PA view, it seems the contrast of the ribs is reduced. This trade-off should be discussed to clarify its implications.

Figure 5a: Consider flipping the images to correct their orientation. Also, to enhance clarity, mark the three regions in distinct colors and consider reordering the images as 2, 1, 4, 5, 6, 3.

Figures 5b and 5d: Maintain a consistent order for PA and AP views across all subfigures for coherence.

**Justification:**

The idea is original and, to my knowledge, the first to explore this direction. The paper is clearly written, well-structured, and easy to follow.

**Reproducibility:**

Sufficient amount of details available for reproducing the main results, but open access is not provided to source code and/or data

**Strengths:**

This appears to be the first paper to explore the potential of using scatter information for reconstruction outside the collimated field of view. The idea is novel and well-motivated, and the paper is supported by a solid experimental evaluation. Overall, the manuscript is clearly written, well-structured, and easy to follow.

**Summary:**

The paper explores the use of scattering signals in the shielded regions of collimated fluoroscopy images to detect and visualize anatomical landmarks and surgical devices. To enhance visibility in these areas, the authors propose applying local histogram equalization. They conduct three separate experiments to investigate relevant factors influencing this approach. The authors conclude that their method shows potential to mitigate field-of-view limitations inherent in collimated imaging, without introducing additional radiation exposure to the patient.

**Weaknesses:**

While the concept is interesting, the proposed method is quite basic. Applying histogram equalization is a technique already widely used in fluoroscopic imaging. At a minimum, the use of CLAHE (Contrast Limited Adaptive Histogram Equalization)  and how to handle the different contrasts of the FOV and outside the FOV should have been described and discussed in more detail.

Although the introduction and related work section is clearly written and accessible, with only six references the coverage of relevant prior work may be incomplete. A broader contextualization within the existing literature would strengthen the paper.

While I understand that this is a proof-of-concept study, several important limitations were not sufficiently discussed. For example, the effects of patient or instrument motion during image averaging could significantly impact results. Additionally, the authors do not address how more advanced techniques (such as deep learning-based denoising or reconstruction methods) might compare to or complement their approach.

Real patient data with real experiments would have been nice to complete the proof-of-concept.

---

### Official Review · Reviewer_bkRC · 2025-07-08

**Recommendation:** 3
**Confidence:** 3

**Clarity:**

The paper is clear and well-written, with minor areas for improvement in clarity

**Feedback:**

My Suggestions:
Improve Citation Practice: Support all factual statements and device specifications with appropriate references. Conduct a deeper literature review, especially in the introduction, and cite recent advances and patents in adaptive collimation and dose reduction. Further, Discuss alternative dose reduction methods beyond collimation, such as reducing the number of images, pulsed fluoroscopy, and adaptive collimation techniques. (i.e. https://doi.org/10.1007/s11548-023-02917-y or Flexman ML, Kruecker J, Panse AS, Toporek GA. Adaptive collimation for interventional x-ray. US20240298995A1. 2022. https://patents.google.com/patent/US20240298995A1)

Explain Physics: Provide a clearer explanation of the physical mechanisms involved (e.g., wave nature of X-rays, scattering, off-focal spot radiation). This is to the reader that might not have sufficient background to understand the underlying phenomenon.

Experimental Verification: Investigate and rule out alternative explanations for observed asymmetries (e.g., rotate the X-ray tube, test different room configurations, or use multiple detector models).

Denoising Approaches: Incorporate and compare denoising techniques (e.g., Laplace, Wiener, diffusion, flow matching, neural networks) to improve the quality of reconstructed images.

Clarify Dose Implications: Clearly state whether frame averaging uses new exposures (increasing dose) or overlays previously acquired frames.

Explicit Contribution: Clearly articulate the main contribution and how it advances the field relative to prior work.

**Justification:**

The paper introduces an interesting and potentially impactful idea, but falls short in key areas. Insufficient citations and a limited literature review weaken its scholarly rigor. Methodological questions—such as the source of observed asymmetries and the role of denoising—remain unresolved. The main contribution is not clearly distinguished from previous work, and some experimental claims lack adequate support. Improvements in citation practice and experimental verification are needed before acceptance; a weak reject is justified in its current form.

**Reproducibility:**

Some amount of details available for reproducing the main results, and open access details are unclear

**Strengths:**

Novelty: Explores a largely overlooked phenomenon—detectable information outside the collimated region in X-ray imaging—and proposes its potential clinical utility.

Experimental Rigor: Employs a variety of experimental setups, including anthropomorphic phantoms and medical devices, to systematically evaluate the phenomenon.

Clinical Relevance: Addresses a real-world challenge in minimally invasive surgery: balancing radiation dose reduction with the need for comprehensive scene awareness.

Potential for Impact: Suggests a method to extract additional diagnostic value from existing radiation exposure, which could improve safety and outcomes in interventional procedures.

Clear Motivation: Articulates the trade-off between dose reduction and field-of-view limitations, which is central to fluoroscopic imaging.

**Summary:**

This paper explores how residual radiation outside the collimated region in fluoroscopic X-ray imaging can reveal anatomical and device information due to physical effects like scatter and off-focal spot radiation. Using phantom and guidewire experiments, the authors show that this unintended information could enhance surgical awareness without extra radiation exposure. The study evaluates the phenomenon’s origins, imaging parameters, and clinical implications.

**Weaknesses:**

Citation Practice and Literature Review: The introduction lacks adequate citations, especially for key statements about collimation and device properties. Several factual claims (e.g., radio-intransparency of Loop-X shutters) are uncited. The paper does not sufficiently discuss alternative radiation reduction strategies (e.g., reducing the number of images, adaptive collimation, or last image hold), nor does it reference recent or relevant literature in these areas.

Clarity and Explanation: Some statements are vague or insufficiently explained, such as the underlying physics responsible for the observed phenomenon (e.g., wave nature of X-rays, specific scattering mechanisms). The paper could more clearly define what the observed phenomenon is in the last sentence of the introduction for better readability and avoiding ambiguity.

Experimental Controls: Observations about asymmetry in irradiation patterns are not thoroughly investigated—possible confounders like detector flaws, shutter placement, or room reflections are not ruled out. No verification (e.g., tube rotation, shutter rotation or detector rotation) is performed to pinpoint the cause.

Contribution Unclear: The core contribution—whether it is the evaluation of collimation patterns—is not clearly enough articulated.

The authors do not attempt any more sophistcated denoising strategies, despite the noisy nature of the detected signals. Both classical and modern denoising methods are omitted.

It is unclear whether the proposal to average multiple frames would significantly increase patient dose, or if only previously acquired frames are used.

The reconstructed images appear significantly different from those in other publications, raising questions about reproducibility or methodological differences. Also, here, a clear reference could provide the required background and verification.

---

### Official Review · Reviewer_RrjJ · 2025-07-09

**Recommendation:** 2
**Confidence:** 4

**Clarity:**

The paper has significant clarity issues that hinder understanding, substantial revision is required to improve clarity

**Feedback:**

I think the paper needs to clarify what the main contribution of the paper is. Is it just the observation that collimated experiments results in some residual reconstruction of the neighboring regions ? If the observation is the only contribution, then its weak contribution. If the authors performed some computational tricks to align the images and increase contrast that are not part of the uncollimated reconstruction pipeline, this needs to be explained properly with side by side results for collimated experiments (with and without the methods employed by the authors)

**Justification:**

The paper is quite interesting, but it does not perform the experiments with the scientific rigor to be presented as novel.

**Reproducibility:**

Sufficient amount of details available for reproducing the main results, and open access is provided (or promised upon acceptance) to source code and/or data

**Strengths:**

1. The paper is based out of a curious observation and ideas which can make use of otherwise wasteful scattered radiation. In my opinion, this is quite clever.
2. This paper also addresses the tradeoff between field of view and radiation exposure, which is a very hard tradeoff to overcome in practice. As such, any approach that can acheive both - wider field of view without increasing radiation exposure is of clinical importance
3. Simple approach and explanation - The authors provide clear and understandable explanation of the phenomenon as well as widely used histogram based methods like CLAHE

**Summary:**

This paper is based on the fact that in collimated fluoroscopy residual radiation outside the collimated region are also illuminated because of scattering of the x-rays. The authors suggest a clever use of these unintended exposure to image neighboring regions around the collimated region. In clinical practice, this can provide a larger field of view without exposing the subject with unnecessary radiation. The paper provides a study of feasibility and provides a proof of concept results for the same.

**Weaknesses:**

1. Motivation: The authors perform experiments with different size and orientation of the collimator (without phantom), but its unclear what was the conclusion of these experiments or why they are necessary in the context of reconstruction. If these experiments were performed to understand the apparatus and detector flaws - it was not discussed in the paper. If it was performed to get some statistical parameters for intensity and deformation to be used for calibration, it not very clearly mentioned.
2. Novelty: Its unclear if this paper is just an observation or do they propose a method of reconstruction as well. In other words, do the collimated reconstruction in current devices generate the residual images or did the authors have to perform some special image enhancement and deformation matching to increase the field of view to the desired contrast. Is this a mere observation of the images currently generated by the collimated x-ray imaging setup ?